# Thioporidiols A and B: Two New Sulfur Compounds Discovered by Molybdenum-Catalyzed Oxidation Screening from *Trichoderma polypori* FKI-7382

**DOI:** 10.3390/antibiotics9050236

**Published:** 2020-05-07

**Authors:** Hirotaka Matsuo, Yoshihiko Noguchi, Rei Miyano, Mayuka Higo, Kenichi Nonaka, Toshiaki Sunazuka, Yōko Takahashi, Satoshi Ōmura, Takuji Nakashima

**Affiliations:** 1Kitasato Institute for Life Sciences, Kitasato University, Tokyo 108-8641, Japan; ynoguchi@lisci.kitasato-u.ac.jp (Y.N.); di17004@st.kitasato-u.ac.jp (R.M.); mayuka@lisci.kitasato-u.ac.jp (M.H.); ken@lisci.kitasato-u.ac.jp (K.N.); sunazukatoshiaki@yahoo.co.jp (T.S.); ytakaha@lisci.kitasato-u.ac.jp (Y.T.); omuras@insti.kitasato-u.ac.jp (S.Ō.); 2Graduate School of Infection Control Sciences, Kitasato University, Tokyo 108-8641, Japan

**Keywords:** anti-microbial activity, MoS-screening, *N*-acetylcysteine, sulfur compound, *Trichoderma polypori*, thioporidiol A and B

## Abstract

Two new sulfur compounds, designated thioporidiol A (**1**) and B (**2**), were discovered by the MoS-screening program from a culture broth of *Trichoderma*
*polypori* FKI-7382. The structures of **1** and **2** were determined as C13 lipid structures with an *N*-acetylcysteine moiety. The relative configuration at the C-5 and C-6 position of **1** was determined by the derivatives of α-methoxy-α-phenylacetic acid diesters, and the absolute configuration of the *N*-acetylcysteine moiety was determined by advanced Marfey’s analysis. Compounds **1** and **2** were evaluated for anti-microbial, cytotoxic and anti-malarial activities. Compound **2** exhibited anti-microbial activity against *Candida albicans* ATCC 64548.

## 1. Introduction

Natural products have been utilized as drugs and agricultural and chemical reagents. Newman and Cragg reported that 64% of metabolites approved as new drugs between 1981 and 2010 were directly or indirectly associated with natural products [1]. Furthermore, according to the KEGG MEDICUS (https://www.kegg.jp/kegg/medicus/) database, which presents information on commercially available medicines [2], 87% of low molecular-weight (0.1–1 kDa) medicines contain nitrogen atoms and 25% of medicines contain sulfur atoms. Therefore, searching natural resources for nitrogen- and sulfur-containing metabolites is expected to yield many unique medicines. 

The search for new medicines is often focused on screening for nitrogen compounds, since the methods used to screen for nitrogen compounds are quite simple. For example, Dragendorff’s reaction can identify tertiary or quaternary amines, and compounds with odd molecular weights can be presumed to have at least one nitrogen atom (the nitrogen rule) [3,4]. However, there have been no reports about simple methods of searching for sulfur compounds. Therefore, we established MoS-screening, a method for screening of sulfur compounds using a combination of molybdenum-catalyzed oxidation and liquid chromatography-mass spectrometry (LC/MS) [5].

The MoS-screening approach allows us to identify compounds containing sulfide in its structure from microbial broths. If sulfur compounds are contained in the broths, they will be oxygenated by Mo-catalyzed oxidation. This oxidation is generally dominant over olefin epoxidation, primary alcohol and secondary alcohol oxidation. In other words, the sulfur compounds in microbial broth could be identified as sulfinyl, sulfonyl or both products, using LC/MS analysis. Then, when the original broth is compared with the oxidative broth, the oxidized peak can be easily identified. Identification of the corresponding oxidative products is relatively straightforward, because the ultraviolet (UV) spectrum is unchanged by oxidation.

In this study, to discover new sulfur compounds from microorganisms, we employed MoS-screening in combination with in-house databases and natural product databases such as the Dictionary of Natural Products (http://dnp.chemnetbase.com/). During our recent MoS-screening, a fungal metabolite with a mass-to-charge ratio (*m/z*) of 372.3221 [M + H]^+^ was deduced to be a new sulfur compound. The producing strain FKI-7382 was isolated from a sediment sample collected at Omuta city, Fukuoka Prefecture, Japan, and identified as the genus *Trichoderma polypori* by DNA barcoding. As a result of purification guided by LC/MS analyses from the culture broth of the fungal strain FKI-7382, a new sulfur compound, designated thioporidiol A (**1**), was isolated together with its analog, thioporidiol B (**2**). Here, we report the fermentation, isolation, structure elucidation and biological activity of **1** and **2**.

## 2. Results and Discussion

### 2.1. MoS-Screening of the Culture Broth of the Fungal Strain FKI-7382

The MoS-screening program was applied to screen microbial broths for new sulfur compounds. Microbial broths were prepared in 50% aqueous ethanol and dispensed across two 96-well plates. (NH_4_)_6_Mo_7_O_24_·4H_2_O and 30% H_2_O_2_ were added to the wells in one plate, while, as a control, only H_2_O was added to the wells of the other plate. After 6 h of shaking, all of the wells were analyzed by LC/MS and the data were compared between the two plates to identify any sulfur compounds. MoS-screening of the broth cultures from 150 different fungal strains yielded a single potential sulfur compound. The candidate compound (**1**) was produced by the fungal strain *Trichoderma polypori* FKI-7382 and showed a retention time of 8.82 min and UV absorbance peaks at 232 nm (Figure 1). High resolution electrospray ionization mass spectrometry (HRESIMS) data show an [M+H]^+^ ion at *m/z* 372.1835, indicating a molecular formula of C_18_H_29_NO_5_S (calculated value for *m/z* 372.1836). The red chromatogram in Figure 1 further indicated that the candidate compound was oxygenated to a sulfonyl (7.95 min, *m/z* = 404.1745 [M+H]^+^). Comparisons of the LC/MS data and UV spectrum of the candidate compound with those of known natural products contained in the Dictionary of Natural Products database (http://dnp.chemnetbase.com/) confirmed that the candidate was a new sulfur compound.

### 2.2. Structure Elucidation of **1** and **2**

Compound **1** was isolated as a colorless amorphous solid by silica gel and octadecylsilyl (ODS) column chromatography from a culture broth of the fungal strain FKI-7382. The ^1^H nuclear magnetic resonance (NMR) and heteronuclear multiple quantum correlation (HMQC) spectra in CD_3_OD (Table 1) indicated the six olefinic protons, two oxymethine protons, five methylene protons, a methine proton adjacent to a heteroatom, an acetyl proton and a methyl proton. The ^13^C NMR spectrum indicated the presence of two carbonyl carbons, six unsaturated carbons, two oxygenated carbons, three carbons seemingly adjacent to a hetero atom and two methyl carbons. The gross structure of **1** was deduced from detailed analyses of 2D NMR data, including ^1^H-^1^H correlation spectroscopy (COSY), HMQC and heteronuclear multiple bond correlation (HMBC) spectra in CD_3_OD (Appendix A). The ^1^H–^1^H COSY spectra revealed the presence of five partial structures, **a**–**e**, as shown in Figure 2A. The connectivity of partial structures **a** and **b** was determined by the HMBC cross-peaks of H-1 to C-3, H-2 to C-4, H-3 to C-1 and H-4 to C-2. The connectivity of partial structures **c** and **d** was determined by the HMBC cross-peaks of H-9 to C-11, H-10 to C-12, H-11 to C-9 and H-12 to C-10. The HMBC cross-peaks of H-6 to C-7 and H-7 to C-6 and their chemical shifts suggested the connectivity of the C-6 and C-7 constituting 1,2-diol. The partial structure **e**, the HMBC cross-peaks of H-1´ to C-3´, H-2´ to C-3´, and 5´-Me to C-4´ and their chemical shifts suggested an *N*-acetylcysteine moiety. Finally, the connectivity of C-13 and C-1´ via a sulfur atom was determined by the HMBC cross-peaks of H-13 to C-1´ and H-1´ to C-13. The geometry of three olefins was determined by detailed analysis of the homo-decoupling of ^1^H-NMR (Table 1) and nuclear overhauser effect (NOE). Thus, the planar structure of **1** was determined as shown in Figure 2A.

Compound **1** has three chiral carbons at C-6, C-7 and C-2´. Freire et al. reported that the absolute or relative configuration of secondary/secondary (*sec*,*sec*)-1,2-diols can be determined by comparing the NMR spectra of the resulting product of α-methoxy-α-phenylacetic acid (MPA) [6]. If the relative stereochemistry of the diol is *anti*, its absolute configuration can be directly determined from the differences of ^1^H-NMR chemical shifts between room temperature and low temperature (∆*δ^T1T2^*) for substituents of the diols. Meanwhile, if the diol is *syn*, the assignment of its absolute configuration requires the preparation of both the bis-(*R*)- and bis-(*S*)-MPA esters, comparison of their room-temperature ^1^H- NMR spectra and calculation of the ∆*δ^RS^* signs for the methines of the α-protons of the diols.

The preparation of the MPA diester of **1** is described below. To protect C-2´ carboxylic acid, compound **1** was derivatized by TMS-diazomethane to methyl ester **3** (Figure 3). The *sec*,*sec*-1,2-diols at C-6 and C-7 of **3** were derivatized to (*R*)-MPA diester **4**. After separation of the crude product by preparative thin layer chromatography (TLC), final purification was conducted by preparative high-performance liquid chromatography (HPLC) to afford **4**. Compound **4** was assigned by 1D and 2D NMR data. ^1^H NMR data measured in 213K and analysis of ∆*δ^T1T2^* values confirmed the *anti*-stereochemistry for C-6 and C-7 (Figure 3, Appendix A). Finally, to determine the absolute configuration of the *N*-acetylcysteine moiety, **1** was defined by advanced Marfey’s analysis of the hydrolysate of **1** after desulfurization with Raney nickel as a catalyst [7,8]. After treatment of **1** with Raney nickel for desulfurization, the product was hydrolyzed by 6 M HCl. The advanced Marfey’s procedure of the hydrolysate led to identification of the stereochemistry of the alanine as the L-alanine (Figure 4).

Compound **2** was isolated as a colorless amorphous solid and determined to have the molecular formula C_18_H_29_NO_5_S by HRESIMS (*m/z* 372.1837 [M+H]^+^, calculated for 372.1836). The ^1^H NMR, ^13^C NMR and UV spectra were similar to that of **1**. These results suggested that **2** is an analog of **1**. The gross structure of **2** was deduced from detailed analyses of 2D NMR data, including ^1^H-^1^H COSY, HMQC and HMBC spectra in CD_3_OD (Figure 2B, Appendix A). These results suggested that **2** is an enantiomer of **1**. Unfortunately, the stereochemistry of compound **2** cannot be determined at C-6, C-7 and the *N*-acetylcysteine moiety due to low yield. 

Compound **1** and **2** were evaluated for several biological activities, including anti-cancer, anti-microbial and anti-malarial activities. As a result of these assays, **1** showed no activity. However, **2** showed anti-microbial activity against *Candida albicans* ATCC 64548, that is a sensitive stain for fluconazole [9]. When the strain ATCC 64548 was treated with 30 µg of **2** on a 6 mm paper disc, an 8 mm inhibition zone was produced. These results suggested that the stereochemistry is important for anti-microbial activity against *C. albicans* ATCC 64548.

## 3. Materials and Methods

### 3.1. General Experimental Procedures

Silica gel and octa-decanoyl-silicon (ODS) were purchased from Fuji Silysia Chemical (Aichi, Japan). All solvents were purchased from Kanto Chemical (Tokyo).

High-resolution electrospray ionization (HRESI)-MS spectra were measured using an AB Sciex TripleTOF 5600+ System (AB Sciex, Framingham, MA, USA). Nuclear magnetic resonance (NMR) spectra were measured using a JEOL JNM-ECA 500 spectrometer (JEOL, Tokyo), with ^1^H-NMR at 500 MHz and ^13^C NMR at 100 MHz in methanol-*d*_4_ (CD_3_OD) and chloroform-*d* (CDCl_3_). The chemical shifts are expressed in parts per million (ppm) and are referenced to residual C*H*D_2_OD (3.31 ppm) and C*H*Cl_3_ (7.26 ppm) in the ^1^H-NMR spectra and CD_3_OD (49.0 ppm) and CDCl_3_ (77.0 ppm) in the ^13^C-NMR spectra. UV spectra were measured with a Hitachi U-2810 spectrophotometer (Hitachi, Tokyo). Infrared radiation (IR) spectra (KBr) were taken on a JASCO FT/IR-4600 Fourier Transform Infrared Spectrometer (JASCO, Tokyo). Optical rotations were measured on a JASCO model DIP-1000 polarimeter (JASCO, Tokyo).

### 3.2. Fermentation of Strain FKI-7382 and Isolation of **1** and **2**

Strain FKI-7382 was grown on a modified Miura’s medium (LcA: consisting of 0.1% glycerol, 0.08% KH_2_PO_4_, 0.02% K_2_HPO_4_, 0.02% MgSO_4_·7H_2_O, 0.02% KCl, 0.2% NaNO_3_, 0.02% yeast extract and 1.5% agar (adjusted to pH 6.0 before sterilization)) slant. A loop of spores of the strain was inoculated into five 500 mL Erlenmeyer flasks containing 100 mL seed medium consisting of 2% glucose, 0.2% yeast extract, 0.5% hipolypeptone, 0.1% KH_2_PO_4_, 0.05% MgSO_4_·7H_2_O and 0.1% agar, which was shaken at 210 rpm on a rotary shaker at 27 °C for 3 days. Twenty bags, each containing 500 g of rice (Hanamasa, Tokyo), 5 g of seaweed tea (ITO EN, Tokyo) and 200 mL of tap water were sterilized, and then a 25 mL aliquot of the seed culture was added to each. The bags were allowed to ferment for 13 days at 25 °C. The total volume of cultured rice (10 kg) was extracted with 10 L of MeOH and centrifuged to separate the cells and supernatant.

The MeOH in supernatant (10 L) was concentrated in vacuo and the suspension was separated by ODS (Chromatorex ODS-DM1020MT) (ϕ100 × 300 mm) middle phase liquid chromatography (MPLC) with a stepwise elution system (0%, 20%, 40%, 60%, 80% and 100% MeOHaq containing 0.1% formic acid; flow rate of 40 mL/min). The 80% MeOH fraction containing **1** was separated by silica gel flash column chromatography (ϕ25 x 150 mm) with a stepwise elution system (CHCl_3_:MeOH = 50:1, 25:1, 12:1, 6:1, 3:1, 0:1; flow rate of 20 mL/min). The eluates were collected in eight-fractions by guiding UV detection of 254 nm. The seventh fraction contained **1** and its analog, thioporidiol B (**2**). This fraction (59.3 mg) was purified by reversed-phase preparative HPLC (CAPCELL PAK C18 MG-III column, ϕ20 × 250 mm; OSAKA SODA, Osaka, Japan) using a solvent system of 65% MeOHaq containing 0.1% formic acid to yield **1** (15.4 mg) and **2** (6.3 mg). 

Thioporidiol A (**1**): pale yellow amorphous solid; [α]_D_^23^ = −14.5 (*c* = 0.1, MeOH); IR (KBr) ν_max_ cm^−1^ 3381, 1728, 1623; UV (CH_3_OH) λ_max_ (log ε) 233 (4.26). 

Thioporidiol B (**2**): pale yellow amorphous solid; [α]_D_^25^ = −29.1 (*c* = 0.1, CH_3_OH); IR (KBr) ν_max_ cm^−1^ 3376, 1729, 1302, 1232; UV (CH_3_OH) λ_max_ (log ε) 232 (4.22). The ^1^H- and ^13^C-NMR data in CD_3_OD are shown in Table 1.

### 3.3. Preparation of MPA Diester of **1**


To protect C-2´ carboxylic acid, compound **1** (7.0 mg) dissoleved in a 1:1 mixture (600 μL) of benzen and methaol was treated with TMS-diazomethane (100 μL) (Tokyo Chemical Indastry, Tokyo, Japan). After stirring for 2 hours at room temperature, the solution was evaporated to yield a C-2´ methyl ester analog **3** (7.0 mg). A small amount of **3** (6.0 mg) dissolved in a 1:1 mixture (200 μL) of dry pyridine and methylene chloride was treated with (R)-MPA (26.0 mg) (Tokyo Chemical Indastry), 4-dimethylaminopyridine (20.0 mg) (Tokyo Chemical Indastry) and *N*, *N*´-dicyclohexylcarbodiimide (32.0 mg) (FUJIFILM Wako Pure Chemical Corporation, Osaka, Japan). After stirring overnight at room temperature, the product was purified by preparative TLC (CHCl_3_:MeOH = 9:1) to yield the crude MPA diester of **1** (Rf value 0.70). Further purification by preparative HPLC (CAPCELL PAK C18 MG-III column, ϕ20 × 250 mm; OSAKA SODA, Osaka, Japan) using a solvent system of 80% MeOHaq containing 0.1% formic acid, afforded to pure MPA diester **4** (1.5 mg). 

### 3.4. Absolute Configuration of the N-Acetylcysteine Moiety of **1**


Advanced Marfey’s analyses for acid hydrolysis of a de-sulfurized **1** with Raney nickel were used to determine the absolute configurations of the *N*-acetylcysteine moiety of both. To a stirred solution of **1** (1.1 mg, 2.96 µmol) in MeOH (1.0 mL), Raney Ni (17.9 mg) was added at room temperature. The suspension was then purged with an H_2_ atmosphere under 1 atm. After stirring for 21 h at room temperature, the suspension was filtered through a filter pad and washed with MeOH (1.0 mL × 3) and H_2_O (1.0 mL × 3). MeOH and H_2_O filtrates were concentrated in vacuo. To hydrolyze the acetyl moiety of *N*-acetylcysteine, the H_2_O filtrate, processed as above, was dissolved in 500 μL of 6 M hydrochloric acid (HCl), followed by heat treatment at 100 °C for 12 h. The product was concentrated to dryness in vacuo and the residue was dissolved in 500 μL of H_2_O. Twenty microliters of 1 M NaHCO_3_ and 50 μL of *N**^α^*-(5-fluoro-2,4-dinitrophenyl)-D-leucinamide (D-FDLA) were added to the 50 μL of hydrolysates, followed by incubation at 37 °C for 1 h. The mixtures were neutralized by addition of 20 μL of 1 M HCl and then concentrated to dryness in vacuo. The resultant dried residue was dissolved in 1 mL acetonitrile, followed by passage through a filter. Similarly, the standard L- and D-alanine were derivatized according to the method described above. The D-FDLA derivatives of the hydrolysate and the standard amino acids were subjected to LC/MS analysis at 40 °C using the following gradient program: solvent A, H_2_O with 0.1% formic acid; solvent B, MeOH with 0.1% formic acid; linear gradient 5–100% of B from 2 to 10 min. 

### 3.5. Anti-Microbial Activity of **1** and **2**

The antimicrobial activities of **1** and **2** against six microorganisms, *Bacillus subtilis* ATCC6633, *Kocuria rhizophila* ATCC9341, *Staphylococcus aureus* ATCC6538P, *Escherichia coli* NIHJ, *Xanthomonas oryzae* pv. *oryzae* KB88 and *Candida albicans* ATCC 64548, were evaluated using the paper disc method. Agar plates were spread with the six strains, and then paper discs including **1** and **2** (final concentration: 1, 3, 10, 30 µg/disc) were placed. All microorganisms, except *X. oryzae*, were incubated at 37 °C. *X. oryzae* was incubated at 27 °C for 48 h. After incubation, the inhibition zones were measured. 

## 4. Conclusions

This study describes new sulfur compounds containing an *N*-acetylcysteine moiety, thioporidiol A (**1**) and B (**2**), produced by the fungal strain *Trichoderma polypori* FKI-7382. To the best of our knowledge, this is the first report describing a natural product containing an *N*-acetylcysteine moiety from fungal metabolites. For example, in the case of nanaomycin H, it is known that an *N*-acetylcysteine moiety is introduced to secondary metabolites as a part of mycothiol [10]. Mycothiol, which is comprised of *N*-acetylcysteine amide-linked to α-glucosamine and *myo*-inositol moieties, is a thiol metabolite that is produced in most actinobacteria and plays a role in protecting against the oxygen toxicity of glutathione in eukaryotes [11]. However, the role and derivation of *N*-acetylcysteine in fungi is still unclear. It is expected that these relationships will be unveiled in the future. The biosynthesis of mycothiol in actinobacteria is accomplished in five steps by several of the enzymes catalyzing the reactions, and the structure is 1-O-[2-[[(2*R*)-2-(acetylamino)-3-mercapto-1-oxopropyl]amino]-2-deoxy-α-D-glucopyranosyl]-D-myo-inositol [12]. The absolute configuration of *N*-acetylcysteine moiety is the same one as in actinobacteria.

## Figures and Tables

**Figure 1 antibiotics-09-00236-f001:**
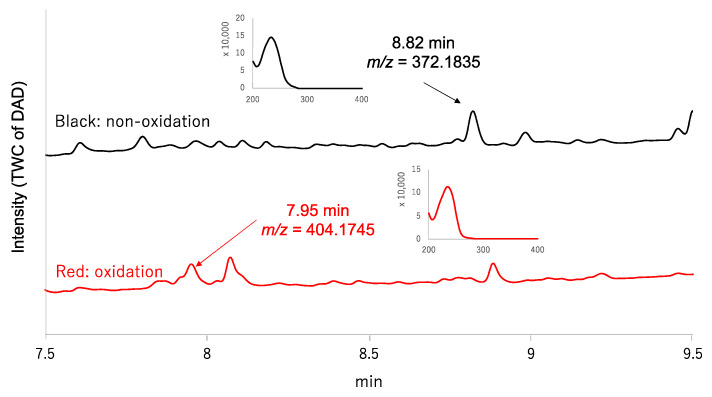
Comparison of total wavelength chromatograms (TWCs) obtained from liquid chromatography-mass spectrometry (LC/MS) analysis of the oxidized and non-oxidized culture broth of the strain FKI-7382.

**Figure 2 antibiotics-09-00236-f002:**
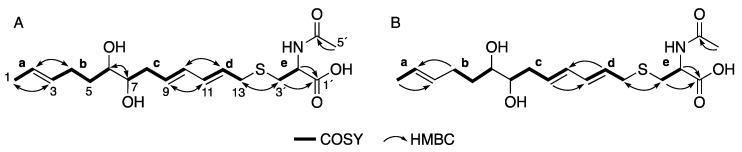
2D NMR correlations of **1** (**A**) and **2** (**B**).

**Figure 3 antibiotics-09-00236-f003:**
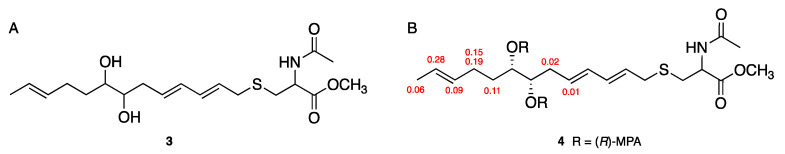
(**A**) The structure of methylated **1**, (**B**) Δδ values (Δδ in ppm) = Δδ*^T1T2^*) obtained for (R)-MPA diester (**4**).

**Figure 4 antibiotics-09-00236-f004:**
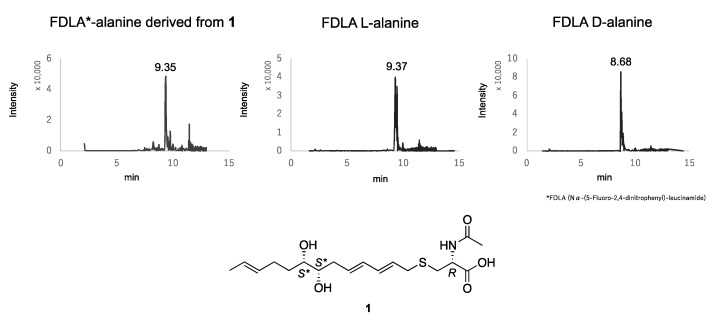
Determination of absolute configuration of the *N*-acetylcysteine moiety derived from **1** by the advanced Marfey’s method. The data are shown as XIC of +TOF MS for *m/z* = 384 to 385.

**Table 1 antibiotics-09-00236-t001:** ^1^H and ^13^C NMR chemical shifts of **1** and **2**.

Thioporidiol A (1)			Thioporidiol B (2)	
Position	*δ* _H_	*δ* _C_	*δ* _H_	*δ* _C_
1	1.63 (dd, 3.5, 1.5, 3H)	18.1	1.63 (dd, 4.0, 1.0, 3H)	18.1
2	5.46 (dd, *J* = 14.0, 3.5 Hz, 1H)	126.0	5.45 (dd, *J* = 14.0, 4.0 Hz, 1H)	126.1
3	5.45 (dddd, *J* = 14.0, 5.0, 5.0, 1.5 Hz, 1H)	132.3	5.45 (dddd, *J* = 14.0, 5.0, 3.0, 1.5 Hz, 1H)	131.1
4	2.03 (dddd, *J* = 14.0, 10.0, 9.0, 5.0 Hz, 1H)	29.9	2.03 (dddd, *J* = 14.0, 8.0, 8.0, 5.0 Hz, 1H)	30.0
	2.20 (dddd, *J* = 14.0, 10.0, 5.0, 5.0 Hz, 1H)		2.15 (dddd, *J* = 14.0, 9.0, 6.0, 3.0 Hz, 1H)	
5	1.41 (dddd, *J* = 14.0, 9.0, 9.0, 5.0 Hz, 1H)	33.6	1.49 (dddd, *J* = 14.0, 8.0, 8.0, 6.0 Hz, 1H)	33.9
	1.68 (dddd, *J* = 14.0, 10.0, 7.0, 3.0 Hz, 1H)		1.56 (dddd, *J* = 14.0, 9.0, 8.0, 4.0 Hz, 1H)	
6	3.38 (ddd, *J* = 9.0, 6.0, 3.0 Hz, 1H)	74.8	3.41 (ddd, *J* = 8.0, 4.0, 4.0 Hz, 1H)	73.9
7	3.40 (ddd, *J* = 8.0, 6.0, 3.5 Hz, 1H)	75.8	3.45 (ddd, *J* = 8.0, 4.5, 4.0 Hz, 1H)	75.0
8	2.19 (ddd, *J* = 14.5, 8.0, 7.0 Hz, 1H)	37.3	2.23 (ddd, *J* = 14.0, 8.0, 7.0 Hz, 1H)	37.5
	2.42 (ddd, *J* = 14.5, 7.0, 3.5 Hz, 1H)		2.34 (ddd, *J* = 14.0, 7.0, 4.5 Hz, 1H)	
9	5.77 (ddd, *J* = 14.0, 7.0, 7.0 Hz, 1H)	132.3	5.74 (ddd, *J* = 14.0, 7.0, 7.0 Hz, 1H)	132.2
10	6.12 (d, *J* = 14.0, 11.0 Hz, 1H)	132.8	6.13 (d, *J* = 14.0, 10.0 Hz, 1H)	132.8
11	6.16 (d, *J* = 14.0, 11.0 Hz, 1H)	134.7	6.17 (d, *J* = 14.0, 10.0 Hz, 1H)	134.6
12	5.53 (ddd, *J* = 14.0, 7.5, 7.0 Hz, 1H)	128.0	5.54 (ddd, *J* = 14.0, 7.5, 7.0 Hz, 1H)	128.1
13	3.19 (dd, *J* = 14.0, 7.0 Hz, 1H)	34.8	3.19 (dd, *J* = 14.5, 7.0 Hz, 1H)	34.8
	3.21 (dd, *J* = 14.0, 7.5 Hz, 1H)		3.21 (dd, *J* = 14.5, 7.5 Hz, 1H)	
1´		173.9		173.8
2´	4.57 (dd, *J* = 8.0, 5.0 Hz, 1H)	53.4	4.57 (dd, *J* = 8.0, 5.0 Hz, 1H)	53.3
3´	2.74 (dd, *J* = 14.0, 8.0 Hz, 1H)	33.0	2.74 (dd, *J* = 14.0, 8.0 Hz, 1H)	33.0
	2.95 (dd, *J* = 14.0, 5.0 Hz, 1H)		2.95 (dd, *J* = 14.0, 5.0 Hz, 1H)	
4´		173.3		173.3
5´	2.00 (s, 3H)	22.4	2.00 (s, 3H)	22.4

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
