# Peer review of "Thioporidiols A and B: Two New Sulfur Compounds Discovered by Molybdenum-Catalyzed Oxidation Screening from Trichoderma polypori FKI-7382"

_antibiotics, 2020, doi:10.3390/antibiotics9050236_

Round 1
Reviewer 1 Report
Please consider the remarks in the attached PDF-file

Author Response
Reviewer 1:
- (1) missing compound numbers whenever a compound is addressed
> Thank you for your kind attention. We added the compound number (line 64).
- these are UV chromatograms from a DAD-UV detector with added information from the MS but not MS chromatograms
> Thank you for your kind attention. We revised a legend of Figure 1 as shown line 73-74.
- 1 and 13 superscript
> We revised it (line 94).
- assignment should be similar in 1 and 2, the carbons can be identified from their HMBC correlations
> We confirmed the HMBC correlations again and revised it (Table 1).
- letters a, b, c ... and numbers too small. letters should not be placed inside the structure
> Thank you for your kind suggestions. We revised it (Figure 2).
- bold numbers
> We revised legend of Figure 2 (line 97).
- no brackets
> We deleted the brackets (line 107-108).
- ??? Greek letters?
> It may be garbled when editing. We revised it (line 119-120).
- A complete structure elucidation is not necessary here, because the data are similar to cmpd 1, see table
> Thank you for your kind suggestion. We deleted the sentence advised by you.
- Bacillus
> We revised it (line 209).
Reviewer 2 Report
In the following article entitled “Thioporidiols A and B: Anti-fungal sulfur compounds discovered by molybdenum-catalyzed oxidation screening from Trichoderma polypori FKI-7382”, the authors describe the discovery by molybdenum-oxidation screening assisted by LCMS and the full characterization of one compound (Thioporidiol A) as well as the partial characterization of a second compound (Thioporidiol B). Overall, the article is perfectly in the scope of Antibiotics and will interest readers with various backgrounds; natural product isolation and structural elucidation. This research article is well written, logical and most of the result/discussion is clear.
Some additions/ clarifications will improve the current version; these are mostly minor points since the research and experiments are well done:
- Abstract: avoid abbreviations such as MPA; prefer a-methoxy-a-phenylacetic acid
- Introduction: L38 “The MoS-screening approach allows us to identify sulfur compounds from microbial broths”. The sulfur atom can exist under various from (thiol, thioether, thioester, disulfide, sulfide, sulfinyl, sulfonyl); it will be nice to add a sentence about this to mention which functional group can be detected with such test.
- Structure elucidation of 1 and 2: the NMR analysis is well presented as well as the conclusion leading to stereoisomers. To improve/ clarify that section, it will be nice to:
- Add the chemical structure of MPA and present the NMR data for 4 (either a zoom of the interesting region in figure 4 or in supplementary information).
- Add FDLA before alanine on figure 5 on top of each chromatograms. Again one chemical structure will help the reader to understand what has been analyzed.
- Is there any explanation for the difference of inhibition between 1 and 2? On P5, L138-142, only 2 is described as active.
- Fermentation of strain FKI-7382 and isolation of 1 and 2: Thioporidiols A and B are relatively hydrophobic compounds; Has chloroform been tried?
- Absolute configuration of the N-acetylcysteine moiety of 1: the methyl ester formation using TMS-diazomethane can be either quickly described here or in supplementary information.
- Anti-microbial activity of 1 and 2: Add some of the relevant inhibition zone in supplementary information with any control; particularly, on those mentioned on P5, L138-142.
- Conclusion: by analogy with mycothiol, any information one the machinery responsible for procuring such compounds? This could help to understand the stereoisomers
Author Response
Reviewer 2:
- Abstract: avoid abbreviations such as MPA; prefer a-methoxy-a-phenylacetic acid
> We revised it (line 15-16).
- Introduction: L38 “The MoS-screening approach allows us to identify sulfur compounds from microbial broths”. The sulfur atom can exist under various from (thiol, thioether, thioester, disulfide, sulfide, sulfinyl, sulfonyl); it will be nice to add a sentence about this to mention which functional group can be detected with such test.
> Thank you for your kind suggestion. At present, we tested only sulfide compounds by MoS-screening. Thus, the sentence about that was added in line 38.
- Structure elucidation of 1 and 2: the NMR analysis is well presented as well as the conclusion leading to stereoisomers. To improve/ clarify that section, it will be nice to:
3.1. Add the chemical structure of MPA and present the NMR data for 4 (either a zoom of the interesting region in figure 4 or in supplementary information).
> Thank you for your kind suggestion. We added the chemical structure of MPA and the 1H NMR data for compound 4 measured by 25 and –30°C in supplementary information (Fig. S13 and 14).
3.2. Add FDLA before alanine on figure 5 on top of each chromatograms. Again one chemical structure will help the reader to understand what has been analyzed.
> Thank you for your kind suggestion. We added the FDLA on each chromatogram (Figure 4).
3.3. Is there any explanation for the difference of inhibition between 1 and 2? On P5, L138-142, only 2 is described as active.
> We added the explanation about activity in line 132-136.
3.4. Fermentation of strain FKI-7382 and isolation of 1 and 2: Thioporidiols A and B are relatively hydrophobic compounds; Has chloroform been tried?
> We have never tried chloroform.
- Absolute configuration of the N-acetylcysteine moiety of 1: the methyl ester formation using TMS-diazomethane can be either quickly described here or in supplementary information.
> We added the methods for derivatization of methyl ester and MPA diester of 1 (Material and method 3.3).
- Anti-microbial activity of 1 and 2: Add some of the relevant inhibition zone in supplementary information with any control; particularly, on those mentioned on P5, L138-142.
> Anti-microbial activity of 1 and 2 was evaluated at concentration 1.0, 3.0, 10 and 30 ug/disc. In this study, a positive control didn't be used.
- Conclusion: by analogy with mycothiol, any information one the machinery responsible for procuring such compounds? This could help to understand the stereoisomers
> The following sentences were added in line 226-229, “The biosynthesis of mycothiol in actinobacteria is accomplished in five step by several of the enzymes catalyzing the reactions, and the structure is 1-O-[2-[[(2R)-2-(acetylamino)-3-mercapto-1-oxopropyl]amino]-2-deoxy-α-D-glucopyranosyl]-D-myo-inositol [11]. The absolute configuration of N-acetylcysteine moiety is same one in actinobacteria.”
Reviewer 3 Report
The article by Matsuo et al. identified two novel sulphur based compounds from Trychoderma polypori, one of which showed to have anti-fungal activities. Sulphur based compounds are not commonly identified, and this study is the first of its kind to screen for sulphur based secondary metabolites from Trychoderma. However, there are some major issues in the article that needs to be addressed. They are as follow:
1) In the title, the author stated that both thioporidiols A and B are anti-fungal sulphur compounds. This is misleading as in the article the authors described only compound 2 as an anti-fungal agent. Hence, the title needs to be changed to better reflect the findings of the article.
2) No results are shown for anti-fungal activity of compound 2 against C. albicans. Please show the result of the disc diffusion assay done to analyze.
3)Line 138, the authors stated that they checked for anti-cancer and anti-malaria activities of the compounds. How were they checked? What assays were used. They are not described in the materials and methods. What were the results of these assays? Please discuss
4) In materials and methods section 3.4, the authors described antimicrobial assays for compounds 1 and 2 against several bacteria. The materials and methods section does not describe the anti-fungal assay against C. albicans. Please explain
5)In the materials and methods please briefly describe the genotype of the C. albicans strain used. Were the disc diffusion assays performed with compounds 1 and 2 compared with a control (For example compared with the zone of inhibition observed with azoles against this strain)? please explain
6)Compound 2 was only found to have anti-fungal properties against C. albicans. Did the authors perform anti-fungal assays on other pathogenic fungi? As testing on one fungus that too only one strain may not justify the anti-fungal activity of compound 2. Similarly compound 1 may show anti-fungal activities to other pathogenic fungi. Please explain.
7)The introduction would greatly benefit if the authors explain the importance of sulphur compounds. The authors only mentioned that sulphur compounds are difficult to isolate. Do the sulphur compounds more advantageous/ efficacious than the nitrogen based compounds?
Minor Issues:
1)Line 46 "in-house databases" the phrase has a mismatched font size?
Author Response
- In the title, the author stated that both thioporidiols A and B are anti-fungal sulphur compounds. This is misleading as in the article the authors described only compound 2 as an anti-fungal agent. Hence, the title needs to be changed to better reflect the findings of the article.
> Thank you for your kind advise. We changed the title to be fit the contents.
- No results are shown for anti-fungal activity of compound 2 againstC. albicans. Please show the result of the disc diffusion assay done to analyze.
> The result of disc diffusion assay for compound 2 was described in line 132-135.
- Line 138, the authors stated that they checked for anti-cancer and anti-malaria activities of the compounds. How were they checked? What assays were used. They are not described in the materials and methods. What were the results of these assays? Please discuss
> We added the methods of anti-cancer and anti-malaria activities in the supplementary information.
- In materials and methods section 3.4, the authors described antimicrobial assays for compounds 1 and 2 against several bacteria. The materials and methods section does not describe the anti-fungal assay againstC. albicans. Please explain
> We misprinted the name of tested microorganisms. Pseudomonas was changed to C. albicans (line 211).
- In the materials and methods please briefly describe the genotype of theC. albicansstrain used. Were the disc diffusion assays performed with compounds 1 and 2 compared with a control (For example compared with the zone of inhibition observed with azoles against this strain)? please explain
> We added more information on Candida albicans ATCC 64548 used (line 133-134). In this time, we simply confirmed antimicrobial and antifungal activities by the disk diffusion assay. We need to investigate the effect of compound 2 against C. albicans because enantiomer compound 1 do not have anti-C. albicans. We will perform with compounds 1 and 2 compared with other antifungal agents.
- Compound 2 was only found to have anti-fungal properties against C. albicans. Did the authors perform anti-fungal assays on other pathogenic fungi? As testing on one fungus that too only one strain may not justify the anti-fungal activity of compound 2. Similarly compound 1 may show anti-fungal activities to other pathogenic fungi. Please explain.
> Indeed, these compounds may show activity against other fungus strain. In this study, we used only C. albicans as fungi. Therefore, we replaced the expression anti-fungal with anti-microbial activity.
- The introduction would greatly benefit if the authors explain the importance of sulphur compounds. The authors only mentioned that sulphur compounds are difficult to isolate. Do the sulphur compounds more advantageous/ efficacious than the nitrogen based compounds?
> We are not discussing the superiority of nitrogen compounds and sulfur compounds as medicine. Both have high ratios among medicines, and it is important to search for these compounds. As far as natural products, the ratio of sulfur compounds (3.6%) are second only to that of nitrogen compounds (24%). There are some simple methods of the screening for nitrogen compounds. However, there is no simple method for detecting sulfur compounds. Therefore, we established MoS-screening method. The presence of heteroatoms results in significant changes in the molecular structure and the number of synthetic methods to afford sulfur-containing molecules is, in practice, restricted to the availability of the appropriate sulfur reagent. Thus, it is considered that natural products containing sulfur at positions difficult to introduce by organic synthesis are useful. If the amount of sulfur-containing natural products increases from now on, it can be expected to be very useful as a drug, like the nitrogen-containing compounds.
Minor Issues:
1)Line 46 "in-house databases" the phrase has a mismatched font size?
> We revised it (line 47).
Round 2
Reviewer 3 Report
The authors have answered satisfactorily to all my queries.